# Development of a Protocol for Anaerobic Preparation and Banking of Fecal Microbiota Transplantation Material: Evaluation of Bacterial Richness in the Cultivated Fraction

**DOI:** 10.3390/microorganisms11122901

**Published:** 2023-12-01

**Authors:** Berta Bosch, Anna Hartikainen, Aki Ronkainen, Filip Scheperjans, Perttu Arkkila, Reetta Satokari

**Affiliations:** 1Human Microbiome Research Program, Faculty of Medicine, University of Helsinki, 00290 Helsinki, Finland; anna.hartikainen@helsinki.fi (A.H.); aki.ronkainen@helsinki.fi (A.R.); 2Department of Neurology, Helsinki University Hospital, 00290 Helsinki, Finland; filip.scheperjans@hus.fi; 3Clinicum, University of Helsinki, 00290 Helsinki, Finland; perttu.arkkila@hus.fi; 4Department of Gastroenterology, Helsinki University Hospital, 00290 Helsinki, Finland

**Keywords:** dysbiosis, inflammatory bowel disease, fecal microbiota transplantation (FMT), *Clostridium difficile*, short-chain fatty acid (SCFA), anaerobic conditions

## Abstract

Fecal microbiota transplantation (FMT) has shown highly variable results in indications beyond recurrent *Clostridioides difficile* infection. Microbiota dysbiosis in many diseases is characterized by the depletion of strictly anaerobic bacteria, which may be crucial for FMT efficacy. We developed a protocol to ensure anaerobic conditions during the entire transplant preparation and banking process, from material collection to administration. The protocol necessitates an anaerobic cabinet, i.e., a non-standard laboratory equipment. We analyzed the population of viable anaerobes by combining cultivation and 16S rRNA gene profiling during the transplant preparation, and after 4, 8, and 12 months of anaerobic or aerobic storage at −80 °C, 78% of fecal species were captured via cultivation. Our findings suggest that strictly anaerobic transplant preparation and storage may preserve species richness better than oxic conditions, but the overall difference was not significant. However, specific anaerobes such as *Neglecta* and *Anaerotruncus* were affected by the oxygen exposure. A storage time of up to 12 months did not affect the presence of cultivated taxa. Noteworthy, our analysis focused on the richness of cultivated anaerobes rather than their abundance, which may have been affected. The benefits of the developed anaerobic protocol in FMT for specific indications remain to be demonstrated in clinical trials.

## 1. Introduction

Microbiota dysbiosis can be defined as the disruption of intestinal bacterial equilibrium and changes in its distribution and metabolic activity [1]. It associates with the pathogenesis of various gastrointestinal disorders, including inflammatory bowel disease (IBD) and colorectal cancer, and systemic ones, like allergy, type 1 and 2 diabetes mellitus, Parkinson’s disease, metabolic syndrome, and obesity [1,2,3,4]. The healthy intestinal microbiota is a population composed of predominantly anaerobic bacteria, of which a large proportion are obligate anaerobes and therefore sensitive to oxygen [5]. Numerous diseases are characterized by the depletion of anaerobic taxa [6].

In IBD, microbiota dysbiosis may play a role in the pathogenesis of chronic mucosal inflammatory lesions [7], and it has been demonstrated that IBD patients have a decrease in specific members of the phylum *Bacteroidetes* and *Firmicutes*, with a substantial increase in *Enterobacteriaceae* and *Enterococcaceae* families [8,9]. Moreover, IBD patients have a reduced abundance of *Faecalibacterium prausnitzii* when compared to healthy controls [7,10]. Further supporting evidence of microbiota dysbiosis is observed in Parkinson’s disease (PD), where the genus *Prevotella* is significantly diminished compared to control subjects [11]. In the context of obesity-related gut microbiota dysbiosis, the enrichment of anaerobic bacteria such as *Allisonella*, *Agathobacter*, *Negativibacillus*, *Dorea*, and *Roseburia*, alongside the depletion of other anaerobic taxa including *Akkermansia*, *Odoribacter*, *Alistipes*, *Bacteroides,* and specifically, *Faecalibacterium prausnitzii* [12], has been linked. In colorectal cancer patients, increased abundances of *Fusobacterium* and enterotoxigenic *Bacteroides fragilis* have been noted. However, more clinical studies are still needed to determine the role of these bacteria in the pathogenesis of colorectal cancer [13]. Observational studies underscore the link between anaerobic *Christensenellaceae* bacteria and well-being, as well as leanness, with a particular increase observed after weight loss [14]. Moreover, *Christensenellaceae* absence has been noted in individuals with Crohn’s disease [15], ulcerative colitis [16], and irritable bowel syndrome [17]. These examples underscore the potential protective role of several anaerobic species in the regulation of intestinal inflammation, metabolism, and neurological functions.

The most efficient currently used treatment to modulate and restore the intestinal microbial environment of humans is fecal microbiota transplantation (FMT), which has been extensively studied for various indications [18,19]. While FMT is a highly effective treatment for recurrent or refractory *Clostridioides difficile* infection (rCDI) with a major rate of recovery (>90%) [20], the results of its use to treat other indications are highly variable [21], specifically concerning ulcerative colitis (UC) [22,23,24]. The different results may be partly due to divergences in the protocols, including different transplant preparation methods. The improvement in UC after the administration of anaerobic FMT material has been associated with an increase in the abundance of *Anaerofilum pentosovorans*, *Bacteroides coprophilus*, and *Clostridium methylpentosum*, all anaerobic species [18,24]. Also, Paramsothy et al. found out that *Eubacterium hallii* and *Roseburia inulivorans* increased in UC responder patients to FMT in comparison with the placebo group [25]. Furthermore, using FMT material prepared anaerobically resulted in a remission rate of 32%, whereas autologous FMT material prepared aerobically used as the placebo resulted in 9% remission [24]. All in all, the anaerobic bacterial population seems to be associated with improved clinical outcomes in UC.

The FMT approach to treat obesity has also been studied [26]. Although in some pilot studies, the balance between benefit and safety is still insufficient, they have shown an improvement in insulin sensitivity and a decrease in chronic low-grade inflammation [27,28]. Clinical improvements when FMT was administered have also been linked to a significant increase in the relative abundance of anaerobic species, like *Clostridium* spp. and *Roseburia intestinalis,* and butyrate-producing bacteria [14], as well as *Akkermansia muciniphila* [29], which has been linked to beneficial metabolic effects [30,31].

Currently, FMT material is usually transported, prepared, and freeze-stored under aerobic conditions, which has resulted in ineffective treatment of rCDI; however, it is recommended that donor feces should be processed as soon as possible, as most fecal bacteria are anaerobic [32]. It has been shown that when FMT samples are handled under aerobic conditions, when suspended, homogenized by blending, and administered, the abundance of obligate anaerobes, such as *Faecalibacterium prausnitzii*, decreases as well as the potential for short-chain fatty acids (SCFA) production, specifically butyrate and acetate [33]. Given the evident decrease observed in obligate anaerobes in UC patients and various other diseases, ensuring anaerobic conditions during FMT material transport, preparation, and delivery might be the key solution to improve the efficacy of FMT in non-rCDI diseases.

In this study, we developed a protocol to ensure anaerobic conditions during the entire transplant preparation and banking process, from material collection to administration. We carried out a microbiological evaluation of the protocol by assessing the population of viable anaerobic bacteria by combining cultivation and 16S rRNA gene profiling during the transplant preparation after 4, 8, and 12 months of anaerobic or aerobic storage at −80 °C.

## 2. Materials and Methods

### 2.1. Fecal Donors

Fecal samples were collected from three healthy adult Finnish donors (D1 = 22-year-old male, D2 = 27-year-old female, and D3 = 54-year-old female) without chronic diseases or prescribed medications and with a BMI between 20 and 24.9, a range considered normal. Each donor gave three fecal samples within one month. In total, 9 samples were used in the study. The donors were interviewed about life habits, chronic diseases, etc., to review their eligibility to act as fecal donors [34] and provided informed consent before sample collection. Laboratory tests for donors included blood tests for hepatitis and human immunodeficiency viruses and stool tests for common enteric pathogens (both bacteria and viruses), antibiotic resistant bacteria and protozoa, helminths, and parasites. Donor testing was performed following international guidelines for fecal donor screening, as described in detail elsewhere [34].

### 2.2. Protocol for Anaerobic Fecal Suspension Preparation and Banking

We developed a practical protocol for the anaerobic transport, processing, and banking of fecal transplants using common household vacuum sealers to seal samples hermetically and protect them from oxygen (Figure 1a). To minimize the oxygen exposure during transportation, donors were instructed to collect and deliver the samples following anaerobic procedures within 1 h after defecation. For that, a written protocol (Appendix A) and an illustrative video (Appendix A) were given to the donors beforehand. Briefly, each stool sample was collected in a 1L plastic, disposable household container and placed inside a household vacuum sealing bag, equipped with an anaerobic gas generator sachet (Thermo Fisher Scientific, Waltham, MA, USA) and an oxygen indicator strip (Merk, Darmstadt, Germany) (Figure 1b). The samples were sealed hermetically with a household vacuum sealer (AirChef 120, Arctixsport, Helsinki, Finland) and the anaerobic package was delivered to the laboratory within two hours and transferred into the anaerobic chamber (10% H_2_, 10% CO_2_, and 80% N_2_; Don Whitley A85 anaerobic workstation, Bingley, UK). The stool was divided into 250 mL containers (Sarstedt 75.9922532, Mawson Lakes, Australia) weighing 30 g of fecal matter. Each aliquot was homogenized with 150 mL of saline solution (NaCl, 9 mg/mL; Baxter, Deerfield, IL, USA) and 20 mL of glycerol (85%; HUS apteekki, Helsinki, Finland) which were poured and mixed with a wooden spatula (Selefa^®^, OneMed, Helsinki, Finland). Aliquots were hermetically closed into sachets by a vacuum sealer inside the anaerobic chamber, then taken out of the chamber and frozen at −80 °C. For administration, the transplants thawed inside the vacuum-sealed bags to preserve the anaerobic conditions and FMT syringes were loaded just before administration to protect the samples from oxygen exposure.

### 2.3. Anaerobic Transplant Preparation and Preservation Study

The outline of the experimental set-up to compare anaerobic and aerobic fecal preparation and preservation is presented in Figure 2. Fecal samples were packed anaerobically and delivered to the laboratory by the donors within 2 h after the sampling, as described above. Once the samples were transferred into the anaerobic chamber, small amounts of feces were sampled into 1.5 mL Eppendorf tubes (Fisher Scientific, Waltham, MA, USA) and frozen at −80 °C for subsequent microbiota analysis. Then, the fecal material was aliquoted into two 250 mL containers, and one aliquot was suspended in saline–glycerol solution in the same ratio/concentration as in the FMT transplant preparations inside the anaerobic cabinet. The sample was divided into several aliquots in 50 mL tubes (VWR^®^ centrifuge tube, 10025-698, Radnor, PA, USA) which were individually packed into hermetically sealed bags inside the anaerobic cabinet and then freeze-stored at −80 °C. These samples are referred to as anaerobic (ANA).

The other 250 mL container was taken out from the anaerobic chamber and left for 2 h under oxygen exposure in a biosafety cabinet (BioWizard Silver line, Kojair^®^, Vilppula, Finland). The exposure time of 2 h was chosen to mimic the time to transport the fecal sample to the laboratory and prepare it for fecal suspensions without anaerobic transport, thus mimicking the exposure time to oxygen in a real-life situation. A sample of feces subjected to oxygen exposure was placed into 1.5 mL Eppendorf tubes and frozen at −80 °C for subsequent microbiota analysis. Next, the 250 mL aerobic (AER) container was suspended in a saline–glycerol solution like the ANA sample and divided into smaller aliquots within 50 mL tubes. Half of the tubes were transferred back to the anaerobic cabinet and sealed hermetically into an anaerobic atmosphere for freeze storage. These samples are referred to as aerobic–anaerobic (AER/ANA). The other aliquots were freeze-stored as such and are referred to as aerobic (AER).

All aliquots were stored in a −80 °C freezer and cultivated at 4-, 8- and 12-month time points. Culturing was conducted also at the baseline from both the anaerobically kept sample and the sample that was exposed to oxygen for 2 h, as described below. Bacterial lawns were collected from densely grown plates (dilution 10^−4^), and the samples were DNA extracted and 16S rRNA amplicons sequenced (see below).

### 2.4. Cultivation of Stool Samples

Cultivation of the fecal suspensions was conducted at 0, 4, 8, and 12 months inside the anaerobic chamber and for the three storage conditions: anaerobic (ANA), aerobic/anaerobic (AER/ANA), and aerobic (AER) (Figure 2). Phosphate-buffered saline (PBS, Medicago AB, SE) was used for serial dilutions, and agar plates were reduced in the anaerobic chamber for 24 h prior to the cultivation. For each condition, stool suspension was diluted in PBS and serial dilutions were performed until 10^−6^. A total of 100 µL of stool dilutions were plated on Gifu Anaerobic Medium (GAM, Nissui Pharmaceutical, Tokyo, Japan), Fastidious Anaerobe Agar (FAA, Tammer BioLab, Tampere, Finland), and Yeast Casitone Fatty Acids (YCFA, [35]) plates, and cultivated for 4 days at +37 °C inside the anaerobic chamber. After incubation, bacterial lawns from the fully grown plates were harvested into 500 µL of lysis RBB buffer [36] for DNA extraction and frozen at −80 °C until further analysis.

### 2.5. DNA Extraction, 16S rRNA Gene Amplicon Sequencing, and Data Analysis

DNA was extracted by using the Repeated Bead Beating (RBB) protocol and the KingFisher Flex 96 (Thermo Fisher Scientific, Vantaa, Finland) and stored at −80 °C until further analysis [37]. DNA concentration was measured using Qubit dsDNA HS Assay kit (Thermo Scientific), Quant-it^TM^ PicoGreen^TM^ dsDNA Assay kit (Thermo Scientific), and Nanodrop^TM^ ND-1000 spectrophotometer (Thermo Scientific) in order to study the purity according to the manufacturer’s instructions.

The DNA of bacterial lawns from the different media (GAM, FAA, and YCFA) from each storage sample were combined in equal amounts to obtain an overall microbiota profile of the cultivated fraction in the sample and subjected to analysis using high throughput 16S rRNA gene amplicon sequencing. In total, 108 bacterial lawn samples and 9 fecal samples together with 5 negative controls (2 blanco samples in the DNA extraction and 3 PCR negative controls) and 1 ZymoBIOMICS microbial community DNA standard (Zymo Research, catalog no. D6306, Irvine, CA, USA) were subjected to 16S rRNA gene profiling.

Microbiota profiling was conducted with 16S rRNA gene amplicon sequencing of the hypervariable region V3–V4 with Illumina MiSeq (Illumina, San Diego, CA, USA) in the Institute of Biotechnology sequencing core facility of the University of Helsinki (FI) according to the previously described protocols [38]. Pre-processing of the sequences was conducted with dada2 pipeline using forward reads [39]. First, adapter sequences were removed with TRIMMOMATIC, and sequences were filtered and trimmed with dada2 function ‘filterAndTrim’ using parameters ‘truncLen = c(237)’, ‘maxN = 0’, ‘maxEE = c(2)’, ‘truncQ = 2’, and ‘trimLeft = c(17)’ [40]. To increase sensitivity to rare sequencing variants, we used the “pseudo-pooling” option in dada2. Taxonomic annotation was conducted with the function ‘assignTaxonomy’ against the RDP training set v18 [41]. Low abundance taxa were filtered, and MicroViz function ‘tax_fix’ was used to fix the phyloseq object [42]. Sequencing data were analyzed with phyloseq and mia [43,44].

Richness estimation was conducted with “pseudo-pooled” data and sub-sampling post-ASV derivation. Bardenhorst et al. suggested using “pooled” data with sub-sampling to consider the connection between sequencing depth and richness [45]. However, we used the option “pseudo-pooling” to minimize computation time and to ensure more sensitive results [45].

The presence/absence table was built with ‘transformCounts’ function in R package mia with a threshold of one sequence, using data that were agglomerated to genus-level. Visualizations were performed with R package ggplot2 [46]. Data have been deposited in ENA under the accession number PRJEB64928.

### 2.6. Statistical Analysis

All figures and tables were generated with RStudio package (R version 4.3.1 for Mac). According to those results, statistical taxonomic differences in the richness between donors, time points, and oxygen exposure levels were calculated with the Wilcoxon signed-rank test. When comparing values, statistical significance was set as *p* below 0.05. Additionally, Fisher’s test was used to analyze the presence and absence of bacterial taxa in the cultivated fractions.

### 2.7. Ethical Considerations

The use of fecal samples from donors for microbiological studies was approved by the Ethics Committee of the Hospital District of Helsinki and Uusimaa Finland (Dnros HUS 124/13/03/01/11 and HUS/1405/2020). The fecal donors provided written informed consent.

## 3. Results

### 3.1. 16S rRNA Sequence Data Analysis

As a result, 16S rRNA gene sequencing of the V3-V4 region generated on average 73,353 (±14,723) reads per sample. Negative controls had a low number of reads (2 to 286 reads). The microbial community DNA standard passed the quality control, and all expected genera were present (Appendix A).

### 3.2. The Prevalent Fecal Microbiota of the Donors

The three donors showed individual microbiota profiles, but they all had *Lachnospiraceae* and *Ruminococcaceae* families and *Faecalibacterium* spp. (Table 1) amongst the dominating bacteria. *Prevotella* and/or *Gemmiger* were among the most prevailing taxa in two out of the three donors. Other highly abundant genera were *Holdemanella* in donor 1, *Dialister* and *Roseburia* in donor 2, and *Blautia* and *Agathobacter* in donor 3.

### 3.3. Preservation of Bacterial Richness in Fecal Suspensions with and without Oxygen Exposure

The effect of time and oxygen exposure levels on cultivated species richness was assessed based on the analysis of observed amplicon sequence variants (ASVs) (Figure 3). As expected, community richness was higher in the donors’ feces as compared to the cultivated bacterial lawns (Figure 3a). The average ASV richness in fecal samples was 264, and in the cultivated fractions of ANA samples, it was 207 at baseline (T_0_), and thus approximately 78% of the bacterial richness could be captured with cultivation on the three different media and subsequent 16S rRNA gene profiling (Figure 3).

Overall, aerobic exposure seemed to result in reduced richness in the cultivated samples, but the difference from the anaerobically prepared material was not statistically significant (Figure 3a). When analyzing the samples from the three individual donors separately, we observed that the initial high richness in the sample may decrease during the storage (donor 1) and that aerobic exposure may decrease the richness of cultivated anaerobic species after 12 months of storage (donor 1 and 3), although the differences were not statistically significant (Figure 3b).

Analysis of the microbiota for each donor’s samples during the 12 months of storage was performed, as microbiota profiles turn out to be donor-specific, as expected.

For donor 1, cultivations at the baseline (0 months) did not show a significant difference in richness between the samples that were treated anaerobically or exposed to oxygen. However, replicates with oxygen exposure showed a larger variance in richness. After 8 months, cultivations showed a richness decrease in ANA and AER/ANA samples when compared to 4 months, but the AER samples did not seem to be affected. However, when comparing the time point 12 months with 4 months, there was a decrease in the richness in all three conditions.

Donor 2 had a lower initial bacterial richness at the baseline (0 months), and the richness from cultivated bacterial lawns was similar between all three conditions, independently of their exposure to oxygen. Donor 2 did not show an overall tendency of decreasing richness over time, but at 4 and 8 months, AER/ANA and AER samples had slightly lower richness (not significant) as compared to the ANA samples. On the other hand, cultivations from the ANA samples showed a wide variance between replicates at 12 months.

Donor 3 had the least richness in microbiota and showed a tendency of lower richness in AER/ANA and AER samples when compared to ANA samples.

Overall, the observations indicate that the individual microbiota of the donor may have an impact on the preservation of bacterial richness during transplant preparation and storage and that there might be also a sample-to-sample variation in how oxygen exposure and storage affect the sample quality.

### 3.4. Presence of Specific Bacterial Genera in the Cultivated Fraction

Subsequently, we examined the occurrence of particular bacterial genera within the cultivated fractions obtained from samples collected during freeze storage and under varying oxygen exposure conditions (Table 2). The listed genera are among the most abundant bacterial taxa found in the donors or genera that are representative of a particular interest due to their association with human health; namely, *Akkermansia*, *Coprococcus*, *Negativibacillus*, and *Dorea*. *Faecalibacterium* spp., *Prevotella* spp., *Roseburia* spp., *Blautia* spp., *Gemmiger* spp., and *Dorea* spp. were present in the cultivated species from all three donors. Overall, oxygen exposure or freeze storage time did not seem to affect their survival, as assessed by their presence in the samples.

*Akkermansia* sp. Was detected in the feces of all three donors via 16S rRNA amplicon sequencing, but it could not be cultivated on GAM, FAA, or YCFA. Therefore, we could not assess its survival after 4, 8, and 12 months nor its aerobic tolerance. *Negativibacillus* spp. and *Holdemanella* spp. were found in donors 1 and 2 via cultivation but not in donor 3. On the contrary, *Coprococcus* sp. Did not grow in the samples of donors 1 and 2 but grew in donor 3’s samples at 0-, 4- and 8-month cultivations. *Dialister* sp. grew when cultivating donor 2 samples at baseline, 4 and 8 months but less frequently from the 12-month time point. Also, in donors 1 and 3 baseline samples, it was present but then showed unstable growth at 4-, 8- and 12-month cultivations.

Overall, the aerobic handling and preparation of fecal samples did not seem to deplete the presence of cultured species strongly and the genera present in the cultivated fraction at the baseline were well preserved in the samples taken during 12 months of storage when compared to the strictly anaerobically treated fecal samples. However, *Dialister* sp. was less frequently recovered in the 12-month samples as compared to the earlier time points, indicating that specific anaerobes may suffer from a long freeze storage time.

### 3.5. Overall Presence and Absence of Taxa in the Cultivated Fraction

The presence/absence of all taxa were analyzed as having over 0.01% relative abundance in the microbiota of the three fecal donors and at least 10% prevalence across the samples. Heat maps showing the presence (grey) or absence (white) of the taxa (Figure 4) showed overall similar patterns between ANA, AER/ANA, and AER samples, indicating good recovery of the anaerobic cultivated taxa after exposure to oxygen and freeze storage. Fisher’s test was used to compare the presence of bacterial taxa between ANA and AER/ANA, or ANA and AER cultivated samples.

Concerning donor 1 samples, an uncharacterized representative of the phylum *Firmicutes* (*p* = 0.014) and *Neglecta* spp. (*p* = 0.039) were less frequently present in the AER samples as compared to ANA, and *Anaeromassilibacillus* sp. Was more prevalent in the AER samples as compared to ANA (*p* = 0.037).

A higher prevalence of *Paludicola* spp. (*p* = 0.036) and lower prevalence of *Anaerotruncus* spp. (*p* = 0.007) were observed for donors 2 and 3, respectively, when comparing AER and ANA samples.

Overall, the impact of two hours of oxygen exposure during the preparation of fecal material does not seem to lead to a major loss of species richness of viable bacteria but may affect specific members of the microbiota. Regarding bacterial lawns used for richness analysis, we collected the lawns from 10^−4^ agar plates, which showed a fully grown lawn from most of the samples and generally did not show less growth after oxygen exposure. However, we did not address the change in total viable counts or that of specific species, genera, or bacterial groups, which would have required a different methodological approach such as selective cultivation, but we presume that aerobic handling could have decreased the viable counts of sensitive anaerobes. Thus, our analysis focused on bacterial richness and the presence of cultivated anaerobes rather than their abundance, and we cannot exclude the possibility that their viable counts could have been affected.

## 4. Discussion

In this study, we developed a protocol to ensure anaerobic conditions during the entire FMT material preparation and banking process, from material collection to administration, and evaluated the impact of long-term storage (12 months) and oxygen exposure on the viable anaerobic bacteria of fecal samples, as determined by cultivation and 16S rRNA gene amplicon sequencing. Stool samples stored at −80 °C for 12 months showed no statistically significant decrease in the bacterial richness of the cultivated fraction, as compared to the baseline. Our results are in line with a recent study showing that freeze storage did not significantly alter viable microbiota composition, although it reduced the overall counts of viable bacteria [33].

Overall, oxygen exposure of two hours reduced the microbiota richness of the cultivated fraction, although not significantly, as compared to the samples kept in anaerobic conditions during the entire procedure. Furthermore, we observed that oxygen exposure affected the presence of certain anaerobic species, namely, *Neglecta* and *Anaerotruncus* spp. as well as uncharacterized representatives of the phylum *Firmicutes*, which were less frequently cultivated from the samples that were exposed to oxygen. While our analysis focused on the overall richness and presence taxa after cultivation and observed in general only a minor impact of oxygen exposure on these characteristics, it should be noted that the abundance of anaerobic species could have been affected. Papanicolas et al. showed that the abundance of obligate anaerobes decreased during the aerobic preparation of feces and that specifically, an important anaerobic species *F. prausnitzii* decreased drastically [33]. Furthermore, the functional potential of the anaerobic population seemed to have suffered as SCFA production of aerobically treated fecal material was lower than that of anaerobically prepared material [33]. In contrast to our results, they also observed a decrease in the overall richness of the viable fraction [33]. The discrepancy in the results may be in part due to different methodologies, as they determined the viable fraction via propidium monoazide (PMA) sample treatment, whereas we used non-selective cultivation to capture viable bacteria.

The importance of ensuring the viability of anaerobic bacteria during fecal preparation has been acknowledged since the beginning of fecal banking. Previously, other studies have introduced protocols for the anaerobic preparation of fecal material, and our study aimed to further refine these protocols and expand the use of anaerobic conditions for material transfer and storage. Hamilton et al. processed fecal material samples in a biological cabinet under N_2_ gas flush prior to freezing at −80 °C [47]. One of the limitations of the Hamilton study was the lack of microbiological evaluation to determine microbiota composition and identify important species of therapeutic interest; therefore, the advantage of N_2_ flushing remains obscure. However, they demonstrated that the efficacy of frozen fecal slurries prepared under N_2_ was comparable to that of freshly prepared fecal suspension in the treatment of rCDI. It still remains to be clarified if anaerobic transport followed by fast fecal preparation and freezing at −80 °C, i.e., without the use of anaerobic cabinet for fecal processing and packaging for banking, could preserve the bacterial richness equally well and thereby simplify the processing.

Some FMT trials have used anaerobically prepared FMT material for the treatment of diseases that are characterized by the depletion of anaerobic bacteria. Costello et al. studied the effect of anaerobic FMT from a fecal donor compared to autologous aerobic FMT [24]. A total of 32% of the participants receiving the pooled donor FMT achieved remission 8 weeks after the treatment compared to 9% when autologous FMT was administered. Remarkably, after 12 months, out of the 42% of participants receiving anaerobic FMT, 32% remained in remission. Additionally, patients that received anaerobically prepared FMT showed an increased abundance of anaerobes such as *Anaerofilum pentosovorans*, *Bacteroides coprophilus,* and *Clostridium methylpentosum*. The latter were associated with a decrease in the total Mayo score, an index that evaluates UC stage, and a disease improvement.

Also, Ding et al. studied the success of the “1-h FMT protocol” on UC after 1 and 3 months of infusion [48]. They treated refractory UC patients with donor FMT that was prepared within 1 h of defecation to decrease oxygen exposure and increase functional microbiota. Although feces were not anaerobically treated, the short processing time was inferred as anaerobically treated samples. Success rates in inducing clinical response were 74% at 1 month after FMT and 51% at 3 months after FMT. Although the Ding study [48] did not compare aerobic and anaerobic FMT material, it seems fair to presume that the clinical success was at least partly due to the implementation of the 1-hour FMT protocol preparation of fecal material and subsequent successful transfer of anaerobic species, considering that microbiota dysbiosis in UC is characterized by the depletion of anaerobes.

We have recently accomplished a clinical trial on Parkinson’s disease using anaerobically prepared FMT material following the protocol presented in this study [49]. The clinical outcomes showed that FMT significantly improved anxiety measures and reduced dopaminergic medication increase over time when compared to the placebo group. Previously, it has been demonstrated that Parkinson’s disease patients have a lower abundance of *Prevotella* spp. compared to healthy controls [11]. The analysis of our anaerobically prepared FMT material showed that *Prevotella* spp. can be cultivated well from the banked feces regardless of oxygen exposure; however, their viable counts were not determined. Thus, it remains debatable whether anaerobic fecal preparation is necessary for the treatment of Parkinson’s disease with FMT, and further clinical studies to compare the clinical efficacy of aerobic and anaerobic FMT would be essential to provide a conclusive response.

The strength of our study is that we developed a practical protocol for keeping anaerobic conditions during the transport and fecal banking by making use of a common household vacuum sealer. The protocol for collecting anaerobic samples can be easily transferred to clinical use. Specimens were closed hermetically with an oxygen scavenger for transport to maintain an anaerobic atmosphere of the fecal slurries after preparation in the anaerobic cabinet. Previously, it has been shown that the collection of samples anaerobically resulted in better preservation of oxygen-sensitive populations when compared to conventional stool sampling, such as *Bifidobacterium bifidum* or *Faecalibacterium prausnitzii* [50], both anaerobes which are of interest also in the context of FMT [51].

The major limitations of our study are that we did not determine the viable counts and focused only on the analysis of bacterial richness and the presence of taxa, and the small sample size might add ambivalence to the results. On the other hand, the strength is that the selected culturing approach using three different media, GAM, FAA, and YCFA, enabled us to cultivate a wide range of gut commensals, representing approximately 78% of the fecal species richness. Thereby, assessing the richness of viable bacteria via cultivation and 16S rRNA gene profiling provided a very good overall estimate, although the non-cultivable fraction constituted approximately 22%, and it was not possible to address the viability of species in this fraction. The bacterial population that did not grow on the chosen media includes *A. muciniphila*, an important health-associated human commensal [52]. *Akkermansia* has selective growth demands and needs a medium containing sugar monomers naturally present in the mucin, such as fucose, galactose, *N*-acetylgalactosamine, and *N*-acetylglucosamine, and its sialic acid (*N*-acetylneuraminic acid) and sulphate variations [53]. Alternatively, Ropot et al. reported that *Akkermansia* can grow in brain–heart infusion and Columbia broth, which were not among our selected cultivation media [53].

Current results from our research revealed that all three donors had a core of predominant bacterial taxa in common. It was comprised of *Lachnospiraceae*, *Faecalibacterium,* and *Ruminococcus*, which were the three most common abundant taxa present in their feces. Also, all three representatives were cultivable and also not affected by the 12-month storage or oxygen exposure. Previous results were in line with the literature, where it is known that human microbiota is comprised of 12 different phyla, of which over 90% belong to *Firmicutes*, *Bacteroidetes*, *Proteobacteria*, and *Actinobacteria* [51]. Amongst these, *Firmicutes* and *Bacteroidetes* dominate the gut microbiota, and their optimal ratio is widely accepted as an indicator of normal intestine homeostasis and related to healthy individuals [54]. *Blautia*, *Coprococcus*, *Dorea*, *Lachnospira*, *Oribacterium*, *Roseburia*, and *L-Ruminococcus* (*Ruminococcus* genus assigned to the *Lachnospiraceae* family) were the main ones detected in the human intestine [55]. Interestingly, *Roseburia* sp. is an obligate anaerobe that is difficult to culture and one of the top twenty most abundant bacteria in the gut microbiome, recognized for its ability to produce butyrate, holding a relative abundance ranging from 5% to 15% [56,57]. Even being considered as a strict anaerobe, *Roseburia* sp. was part of the cultivable fraction after being exposed to oxygen and stored. The two following studies support the need to preserve certain species due to their reduced number of patients and their therapeutic activities or future approaches. Firstly, FMT studies conducted in UC patients showed an increase in *Roseburia* sp. in the remission group and higher SCFA levels than those who did not experience remission [58]. *Roseburia* is also proposed as a candidate species for the probiotic treatment of IBD patients [59]. Secondly, Frank et al. showed that *Lachnospiraceae* and *Bacteroidetes* microbial populations were decreased and *Proteobacteria* families were increased in IBD patient samples when compared to control ones [60].

Our results from the presence of specific taxa in the cultivated fraction after oxygen exposure and during storage showed a fair survival of anaerobic species. *Faecalibacterium* is classified as a strict anaerobe, but it survived oxygen exposure and could be retrieved in the cultivated fraction of all samples. Interestingly, Khan et. al. showed that *Faecalibacterium* cells can survive in oxygenated environments via extracellular electron transfer to oxygen [61]. On the other hand, Papanicolas et al. processed samples in ambient air and concluded that relevant commensal taxa abundance, including *Faecalibacterium prausnitzii*, was reduced [33]. Also, *Prevotella* seems to survive in oxygen exposure regardless of being classified as strictly anaerobic [62]. Aerotolerance in *Prevotella* isolates was observed by Silva et al. after the isolation, during the exponential growth phase, and after 4 months of laboratory handling [62]. Indeed, it seems that although most gut bacteria are categorized as obligate anaerobes, a substantial range of oxygen tolerance exists in the microbiota and its members have developed various strategies to resume growth after exposure to oxygen [63].

The analysis of taxa presence/absence in cultivated fractions revealed a loss of specific species upon oxygen exposure, with donor-dependent variations. In the case of donor 1, an unknown member of phylum *Firmicutes* and *Neglecta* spp. was not cultivable in samples exposed to oxygen for two hours. *Neglecta* sp. was isolated and classified as a strictly anaerobic genus belonging to the order *Clostridiales*, a butyric, acetic, and propionic acid producer [64]. Conversely, for donor 2, *Paludicola* sp. was more frequently recovered after oxygen exposure, while donor 3 showed negative effects on *Anaerotruncus* sp. due to oxygen exposure and storage. *Paludicola* is a novel anaerobic genus belonging to the *Ruminococcaceae* family, whose function in the human gut is not known yet [65]. *Anaerotruncus* sp., an anaerobic, butyrate-producing member of the gut microbiota, has been associated with improving experimental autoimmune encephalomyelitis, in a mouse model for multiple sclerosis, potentially through the induction of RORγt^+^ regulatory T cells with autoimmune-regulating functions [66]. Moreover, when analyzing highly complex microbiota samples, lower abundant bacterial taxa might be not detected and therefore are only found in the cultured fraction but not in the original fecal sample. Examples of the latter are *Enterobacteriaceae* and *Lactobacilli*, taxa that are typically low abundant but might have an advantage when growing in a laboratory with more fastidious bacteria, showing higher relative abundance in cultured samples. It may be speculated that the extent to which oxygen compromises the viability of gut bacteria could be donor-dependent and even sample-specific. Indeed, it has been noted that there is a great variability in the oxygen tolerance of obligately anaerobic bacteria and there might be a need for stringent anaerobiosis when culturing and preserving strains for live biotherapeutic products [67].

## 5. Conclusions

Although we observed an overall excellent preservation of species richness after oxygen exposure, the examples of lost viability for specific species such as *Neglecta* and *Anaerotruncus* spp. still underline the deleterious effect of oxygen on FMT material and the fact that anaerobic preparation could be essential. The protocol presented in this study strives toward maximum protection of fecal material from oxygen during the entire transplant preparation and banking process, from material collection to administration, and it may result in a clinical advantage in treating disorders where microbial imbalance is characterized by the depletion of strictly anaerobic bacteria. In the treatment of patients with recurrent *Clostridioides difficile* infection (rCDI) via FMT, fecal preparation under ambient air has proven to result in excellent treatment outcomes and thus there is no apparent need to consider anaerobic preparation for the treatment of rCDI, the main indication for FMT.

## Figures and Tables

**Figure 1 microorganisms-11-02901-f001:**
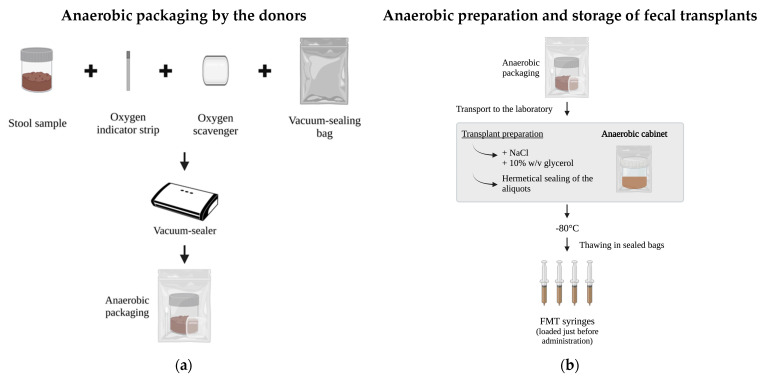
Protocol for the anaerobic preparation of frozen fecal inoculum for fecal microbiota transplantation (FMT) (**a**) from anaerobic packaging by the donors to (**b**) the anaerobic preparation of transplants in the laboratory; donor samples are preserved from oxygen by anaerobic packaging and transported to the laboratory after defecation. A total of 30 g of feces are suspended into 150 mL of sterile saline by mixing with a spatula in a 250 mL screw cap container and 10% glycerol (final concentration) is added inside the anaerobic cabinet. The package is sealed hermetically in a vacuum sealing bag inside the anaerobic chamber and freeze-stored at −80 °C. For FMT administration, samples are allowed to thaw inside the sealed bag at +37 °C or room temperature, mixed briefly, and aliquoted to FMT syringes. If necessary, a pre-sterilized and stainless-steel strainer can be used to remove non-suspended particles present in the suspension before loading the syringes.

**Figure 2 microorganisms-11-02901-f002:**
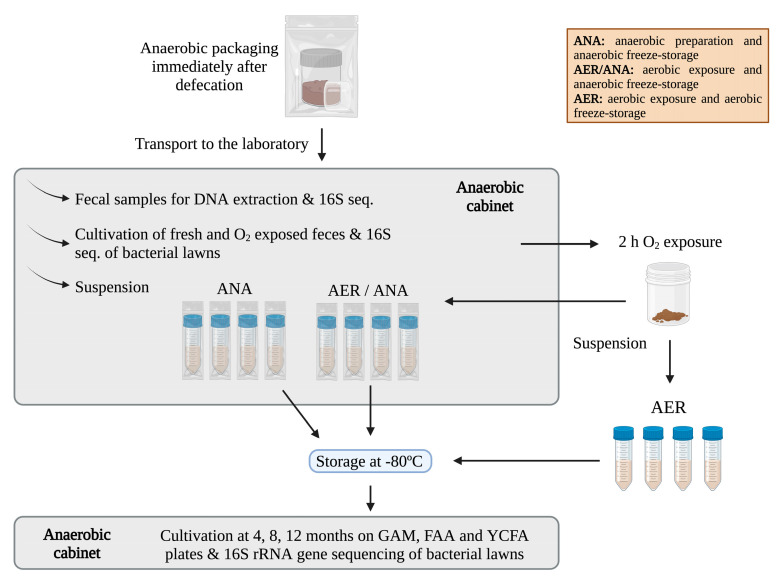
Protocol for the anaerobic preparation and preservation study of frozen fecal inoculum. Donor samples were packed anaerobically and transported to the laboratory. Samples were taken from fresh feces and cultivations were conducted, and further DNA extraction and 16S rRNA sequencing was performed. Feces were prepared into suspensions and for storage using three conditions; anaerobic (ANA), aerobic/anaerobic (AER/ANA), and aerobic (AER), according to their oxygen exposure during preparation and storage at −80 °C, as described in detail in the main text. Cultivations from the freeze-stored samples were conducted at time points of 4, 8, and 12 months on GAM, FAA, and YCFA plates, bacterial lawns were collected, DNA was extracted, and 16S rRNA gene amplicons were sequenced.

**Figure 3 microorganisms-11-02901-f003:**
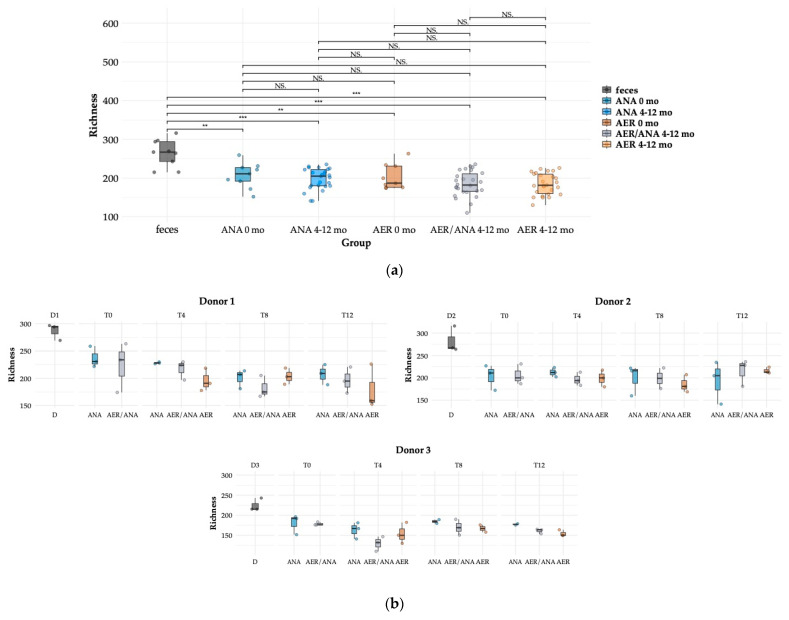
Species richness estimates in the fecal samples from the three healthy fecal donors and the cultivated bacterial lawns after the storage of fecal suspensions: (**a**) all samples from all three donors combined and (**b**) the samples from individual donors. Oxygen exposure degree during the preparation and freeze storage of fecal suspension is indicated as ANA (anaerobic), AER/ANA (aerobic/anaerobic), and AER (aerobic) (see main text for further details). The time points refer to the baseline (T_0_) and 4-, 8- and 12-month storage at −80 °C before cultivation (T_4_, T_8,_ and T_12_). Asterisks indicate a statistically significant difference (NS = not significant, ** = *p* < 0.01, *** = *p* < 0.001) between the samples.

**Figure 4 microorganisms-11-02901-f004:**
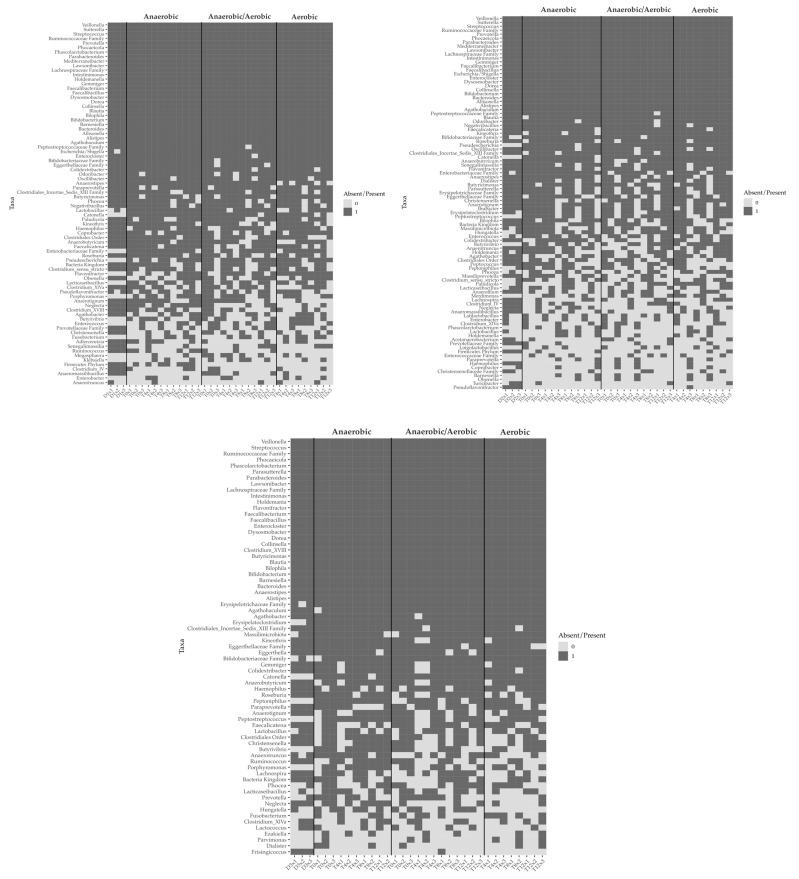
Presence of bacterial taxa in the fecal and cultivated bacterial lawn samples of the three donors. Color gray = present, color white = absent. Dx = donor’s fresh feces, Tx = time point cultivations. All taxa with over 0.01% relative abundance of the total microbiota and over 10% prevalence in all samples are included.

**Table 1 microorganisms-11-02901-t001:** Relative abundance of the six most predominant bacterial taxa in the three donors’ feces. Relative abundance means ± standard deviations (mean ± SD) are expressed based on three replicate samples (*n* = 3).

Predominant Taxa	
Donor 1	Mean ± SD
*Lachnospiraceae*	21.109 ± 4.892
*Faecalibacterium*	15.811 ± 2.078
*Prevotella*	12.076 ± 4.127
*Ruminococcaceae*	5.635 ± 3.823
*Gemmiger*	5.293 ± 0.544
*Holdemanella*	3.716 ± 1.125
Donor 2	
*Faecalibacterium*	19.190 ± 3.868
*Lachnospiraceae*	15.258 ± 2.220
*Ruminococcaceae*	10.280 ± 1.159
*Prevotella*	9.407 ± 5.397
*Dialister*	6.984 ± 1.456
*Roseburia*	4.465 ± 1.558
Donor 3	
*Faecalibacterium*	17.580 ± 4.440
*Lachnospiraceae*	16.911 ± 4.202
*Agathobacter*	13.214 ± 5.314
*Gemmiger*	8.984 ± 0.939
*Ruminococcaceae*	6.956 ± 3.176
*Blautia*	5.953 ± 1.543

**Table 2 microorganisms-11-02901-t002:** The presence of specific taxa in the cultivated fraction of microbiota in the three fecal donors at baseline (T_0_) and after 4 (T_4_), 8 (T_8_), and 12 months (T_12_) of storage at −80 °C, as assessed by the 16S rRNA gene profiling of the grown bacterial lawns. Preparation and preservation conditions are shown as ANA (anaerobic), AER/ANA (aerobic/anaerobic), and AER (aerobic).

Donors/Time Points and Cultivated Bacteria	1			2			3		
**0 months (T_0_)**	**ANA**	**AER/ANA**	**AER**	**ANA**	**AER/ANA**	**AER**	**ANA**	**AER/ANA**	**AER**
*Faecalibacterium*	+	+	+	+	+	+	+	+	+
*Prevotella*	+	+	+	+	+	+	+	+	+
*Akkermansia*	(+)	-	-	-	-	-	-	-	-
*Roseburia*	+	+	+	+	+	+	+	+	+
*Coprococcus*	-	-	-	-	-	-	-	+	+
*Negativibacillus*	+	+	+	+	+	+	-	-	-
*Gemmiger*	+	+	+	+	+	+	+	+	+
*Holdemanella*	+	+	+	-	(+)	(+)	-	-	-
*Dialister*	(+)	(+)	(+)	+	+	+	+	(+)	(+)
*Blautia*	+	+	+	+	+	+	+	+	+
*Dorea*	+	+	+	+	+	+	+	+	+
**4 months (T_4_)**									
*Faecalibacterium*	+	+	+	+	+	+	+	+	+
*Prevotella*	+	+	+	+	+	+	(+)	+	+
*Akkermansia*	-	-	-	-	+	-	-	-	-
*Roseburia*	+	(+)	+	+	+	+	+	(+)	(+)
*Coprococcus*	-	-	-	-	-	-	(+)	-	-
*Negativibacillus*	+	+	+	+	+	+	-	-	-
*Gemmiger*	+	+	+	+	+	+	+	(+)	(+)
*Holdemanella*	+	+	+	(+)	(+)	+	-	-	-
*Dialister*	-	(+)	-	+	+	+	-	-	-
*Blautia*	+	+	+	+	+	+	+	+	+
*Dorea*	+	+	+	+	+	+	+	+	+
**8 months (T_8_)**									
*Faecalibacterium*	+	+	+	+	+	+	+	+	+
*Prevotella*	+	+	+	+	+	+	+	+	(+)
*Akkermansia*	-	-	-	-	-	-	-	-	-
*Roseburia*	+	+	+	+	+	+	+	+	(+)
*Coprococcus*	-	-	-	-	-	-	(+)	(+)	(+)
*Negativibacillus*	+	+	+	+	+	+	-	-	-
*Gemmiger*	+	+	+	+	+	+	+	+	+
*Holdemanella*	+	+	+	(+)	(+)	(+)	-	-	-
*Dialister*	-	-	(+)	+	+	+	(+)	-	(+)
*Blautia*	+	+	+	+	+	+	+	+	+
*Dorea*	+	+	+	+	+	+	+	+	+
**12 months (T_12_)**									
*Faecalibacterium*	+	+	+	+	+	+	+	+	+
*Prevotella*	+	+	+	+	+	+	(+)	(+)	(+)
*Akkermansia*	(+)	-	-	-	+	(+)	-	-	-
*Roseburia*	+	(+)	(+)	+	+	+	+	+	+
*Coprococcus*	-	-	-	-	-	-	-	-	-
*Negativibacillus*	+	+	+	+	+	+	-	-	-
*Gemmiger*	+	+	+	+	+	+	+	+	+
*Holdemanella*	+	+	+	(+)	-	-	-	-	-
*Dialister*	-	-	-	+	+	+	-	-	(+)
*Blautia*	+	+	+	+	+	+	+	+	+
*Dorea*	+	+	+	+	+	+	+	+	+

+ detected in 2 or 3 replicates; (+) detected in 1 replicate; - not detected.

## Data Availability

Publicly available datasets were analyzed in this study. These data can be found in the ENA deposit of sequences with the accession number PRJEB64928.

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
