# Peer review of "Development of a Protocol for Anaerobic Preparation and Banking of Fecal Microbiota Transplantation Material: Evaluation of Bacterial Richness in the Cultivated Fraction"

_microorganisms, 2023, doi:10.3390/microorganisms11122901_

Round 1
Reviewer 1 Report (Previous Reviewer 2)
Comments and Suggestions for Authors
The authors addressed concerns raised towards the original submission. Although the FMT process design includes aerobic exposure of the fecal sample for the time period immediately after collection, it might not be feasible to achieve less exposure for larger population-based studies. As 16S sequencing depth was insufficient to detect LAB and EB in the fecal sample, qPCR could/should have been used to quantify amounts in fecal samples. This approach would allow the reader insight into the degree of bias due to cultivation. What is the point of comparing diversity in fecal with cultivation based samples and make conclusions about distortions associated with oxygen exposure at various steps when the authors rightfully argue that cultivation distorts diversity and 16S has insufficient sequencing depth?
Author Response
Please see the attachment.

Reviewer 2 Report (New Reviewer)
Comments and Suggestions for Authors
The article contains interesting information, but minor revisions are needed:
-There is a lack of references in some sentences.
-Please find attached my comments about the presente article in the PDF with corrections.

-In general, the writing sometimes errs on the side of being more colloquial than scientific. Authors should review this and try to write more scientifically.
Author Response
Please see the attachment.

Reviewer 3 Report (New Reviewer)
Comments and Suggestions for Authors
Dear Authors,
The submitted paper shows the potential of a relevant scientific contribution with practical applicability. Nevertheless, there are issues that need updates before the acceptance for publication. The title should emphasize that the protocol is in the developmental phase. The abstract contains details that are challenging to interpret. The introductory part should present the previous achievements regarding similar protocols and highlight the remaining issues. Besides, it should have a more uniform approach in describing the conditions representing indications for fecal microbiota transplantation (FMT). The donor's lifestyle details would be appreciated. The discussion lacks comparison with the other protocols for similar purposes. Besides, it contains inconsistencies regarding the statistical support for some of the comments. Please find below the detailed comments and suggestions.
Title, abstract, and keywords
*Title*
- Please rephrase into Development of the protocol for anaerobic preparation and banking of fecal microbiota transplantation material: evaluation of bacterial richness in the cultivated fraction.
*Abstract*
- Line 16: In seems more suitable than of.
- Lines 23–4: The interpretation is challenging. If the overall difference is insignificant, is it justifiable to adhere to anaerobic conditions? Aerobic might replace oxidic.
- Lines 26–8: What could be the consequences of the altered abundances? Do they represent the limitations of the developed protocol?
*Keywords*
- Unexplained abbreviations might be avoided as keywords.
Introduction
- Does the literature offer protocols for anaerobic preparation and banking of FMT material? What is the current state of the art and most relevant issues?
- Lines 43–60: The details about dysbiosis features in colorectal cancer would be appreciated.
- Lines 61–89: The info is lacking about FMT efficiency in the other conditions characterized by microbiota dysbiosis, such as Parkinson’s disease or metabolic syndrome. The efficiency in ulcerative colitis (UC) might contain fewer details.
- Line 96: Please explain the abbreviation SCFA.
Materials and Methods
- Lines 108–14: Did the donor requirements refer to smoking, alcohol consumption, and dietary habits? Quoting the main groups of the lab tests would be appreciated.
- Lines 229–30: The total number of samples pointed towards the non-parametric statistics. Therefore, the Shapiro-Wilk test was unnecessary.
Results
- The data format in Table 1 should be median with minimum and maximum.
- Throughout the Manuscript, please provide the precise P-values.
Discussion
- In general, the discussion would benefit from shortening. Besides, it remained elusive whether adherence to the strictly anaerobic conditions should be a general requirement or should depend on the type of the pathologic condition.
- Lines 366–8: If the statistical significance is absent, it is not suitable to report the decrease.
- Lines 434–42: What are the advantages and disadvantages of your protocol compared with the others available in the literature for the same purposes?
- Lines 443–59: Please consider additional efforts to comment on the practical limitations of your protocol, such as those related to the specific equipment and donors' education for collecting the stool samples at home.
Round 2
Reviewer 3 Report (New Reviewer)
Comments and Suggestions for Authors
Dear Authors,
Your efforts in revising the Manuscript are evident and appreciated. The overall quality is at a higher level than the original submission and meets the criteria for acceptance. Congrats for the work.
This manuscript is a resubmission of an earlier submission. The following is a list of the peer review reports and author responses from that submission.
Round 1
Reviewer 1 Report
Comments and Suggestions for Authors
1. The resolution of the picture is generally poor, and the aesthetic degree is insufficient.
2. Please note the formality of the form.
3. The introduction is too wordy, emphasizing the importance of fecal bacteria transplantation and clinical status.
4. The abstract mentioned fecal transplantation for treating C. difficile infection, but the experimental design did not reflect it. What is the significance of emphasizing the treatment of C. difficile infection here?
5. The article mentions Our findings suggesting that strictly anaerobic transplant preparation and storage may preserve species richness better than oxic conditions, but the overall difference wasn’t significant. The small sample size may have contributed to this ambiguity.
6. The experimental design is relatively simple, and the conclusions obtained have limited significance for clinical guidance.
Reviewer 2 Report
Comments and Suggestions for Authors
The authors report on the effects of exposure to oxygen during preparation of FMT samples. Their carefully designed 16S sequencing/cultivation based studies show some minor effects of exposure to oxygen during FMT prep on diversity of surviving bacteria. The reviewer struggled with fully understanding data presented in figure 4, maybe the legend can be expanded to explain Dlx vs Tlx. One would assume that all taxa that were later cultured (Tlx?)were also present in the fecal samples (Dlx?). However, the figure shows quite a few taxa (including Enterobacteriaceae and lactobacillus) that were identified in the cultured samples but not identified in fecal samples. Reasons for this discrepancy should at least be discussed. One design concern is a lacking comparison of the effect of collecting anaerobically vs aerobically as at this stage more of the bacteria are metabolically active and thus susceptible to oxygen exposure. From the data presented here it seems that collecting anaerobically might be more crucial than maintaining anaerobic conditions later in the FMT process, but this should be shown or at least discussed.
Reviewer 3 Report
Comments and Suggestions for Authors
The Authors touched on a very important topic. There are still no strict guidelines on how to prepare and store FMT materials. It is also extremely important to determine the value of such material in terms of the presence and quality of anaerobic bacteria.
However, a few things need to be clarified.
What was the age and BMI of the donors?
It would be good to clearly state how many samples were ultimately analyzed: did each donor provide only one sample?
The Authors concluded that: "our results indicate that the extent to which oxygen compromises the viability of gut bacteria could be donor and even sample specific, although the overall species richness didn’t decrease significantly after two hours of oxygen exposure".
Aren't these conclusions too far-reaching if the research was carried out on samples of only three donors? Is there an experiment planned in which more samples from more patients will be analyzed?
Do the Authors assume in their protocol that each donor will always have this common household vacuum-sealers at their disposal? And that they will want to use this device to pack material that is not very attractive in terms of aesthetics and smell?
A slightly different experiment would be useful, comparing bacterial profiles and the abundance of anaerobic bacteria in stool samples delivered in a package ensuring strictly anaerobic conditions and in "normal" packaging that does not provide such conditions.
The figures in this manuscript are very important. Could they be presented in a better resolution and larger size?